# Unraveling the complexities of Last Glacial Maximum climate: the role of individual boundary conditions and forcings

Xiaoxu Shi[1,2], Martin Werner[2], Hu Yang[1,2], Roberta D'Agostino[3,4], Jiping Liu[5,1], Chaoyuan Yang[1], and Gerrit Lohmann[2]

[1]Southern Marine Science and Engineering Guangdong Laboratory (Zhuhai), Zhuhai, China
[2]Alfred Wegener Institute, Helmholtz Center for Polar and Marine Research, Bremerhaven, Germany
[3]Max-Planck-Institute for Meteorology, Hamburg, Germany
[4]Italian National Research Council, Insitute of Atmospheric and Climate Sciences, via per Monteroni Km 1.2, 73100, Lecce, Italy
[5]School of Atmospheric Sciences, Sun Yat-sen University, Zhuhai, China

**Correspondence:** Xiaoxu Shi (shixiaoxu@sml-zhuhai.cn)

**Abstract.** In order to quantify the relative importance of individual boundary conditions and forcings, including greenhouse gases, ice sheets, and Earth's orbital parameters, on determining Last Glacial Maximum (LGM) climate, we have performed a series of LGM experiments using a state-of-the-art climate model AWI-ESM, in which different combinations of boundary conditions and forcings have been applied following the protocol of Paleoclimate Modelling Intercomparison Project phase 4 (PMIP4). In good agreement with observational proxy records, a general colder and drier climate is simulated in our full-forced LGM experiment as compared to the present-day simulation. Our simulated results from non-full-forced sensitivity simulations reveal that both the greenhouse gases and ice sheets play a major role on defining the anomalous LGM surface temperature compared to today. Decreased greenhouse gases in LGM as compared to present-day leads to a non-uniform global cooling with polar amplification effect. The presence of LGM ice sheets favors a warming over Arctic Ocean and North Atlantic in boreal winter, as well as a cooling over regions with the presence of ice sheets. The former is induced by a strengthening in the Atlantic meridional overturning circulation (AMOC) transporting more heat to high-latitudes, whilst the latter owing to the increased surface albedo and elevation of ice sheets. We find that the Northern Hemisphere monsoon precipitation is influenced by the opposing effects of LGM greenhouse gases and ice sheets. Specifically, the presence of ice sheets leads to significant drying in the Northern Hemisphere monsoon regions, while a reduction in greenhouse gases results in increased monsoon rainfall. Based on our model results, continental ice sheets exert a major control on atmospheric dynamics and the variability of El Niño-Southern Oscillation (ENSO). Moreover, our analysis also implies a nonlinearity in climate response to LGM boundary conditions and forcings.

## 1 Introduction

The Last Glacial Maximum (LGM), approximately 21,000 years before present, was the most recent extremely cold period when the Northern Hemisphere ice sheets reached their maximum size, therefore it has been a focus of the Paleoclimate Modelling Intercomparison Project (PMIP). Compared to today, the LGM climate is characterized by global cooling and

dryness (Clark et al., 2009). A surface cooling of $4.0 \pm 0.8$ K between LGM and today is estimated by synthesizing various proxy data both on land and ocean (Annan and Hargreaves, 2013). By combining an extensive collection of geochemical proxies with isotope-enabled model results, a constraint on global mean LGM cooling of -6.5 to -5.7 K was estimated (Tierney

et al., 2020). Noble gases in groundwater indicated that the low-altitude, low-to-mid-latitude land surface cooled by $5.8 \pm 0.6$ K during the LGM (Seltzer et al., 2021). In a more recent study in which model results and proxy records were blended via a data assimilation approach, a global annual mean surface air temperature anomaly of $-4.5 \pm 0.9$ K relative to today is suggested (Annan et al., 2022). A question that arises here is why was the LGM climate so different from today?

Several factors can affect the temperature changes on regional and global scale. According to the Milankovitch theory, the

Earth's orbital changes are the primarily driver for the past glacial cycles (Clark et al., 2009). The LGM epoch corresponds to a period of relatively low obliquity, which reduces the high-latitude temperature on annual mean perspective. Additionally, during the LGM aphelion took place in the Northern Hemisphere summer, helping to maintain a relatively cool boreal summer (Berger, 1977) and promoting the growth of Northern Hemisphere ice sheets. In addition to the Earth's orbital changes, strong cooling during the LGM could also be induced by the reduced greenhouse gases with respect to the pre-industrial era, as well

as the existence of the Laurentide and Fennoscandia ice sheets which are absent today. The additional land-ice coverage during the LGM contributed to a global mean sea level drop by about 130 m as well as changes in the land–sea distribution (Yokoyama et al., 2000).

Besides the temperature differences, pollen-based precipitation reconstructions indicate global dryness with local wetting during the LGM with respect to present-day (Bartlein et al., 2011). On one hand, changes in greenhouse gases and or-

bital parameters lead to changes in the net energy input (NEI) into the atmospheric column (i.e., the difference in radiative fluxes between the top-of-atmosphere and the Earth's surface), which further modify the atmospheric general circulation and dynamically-driven precipitation (D'Agostino et al., 2019). On the other hand, the presence of the ice sheets over North America and Fennoscandian redirects the low-level winds which has strong influences on moisture transport and regional precipitation (Liakka and Lofverstrom, 2018). Moreover, thermodynamic effects associated with changes in specific humidity can

also play a role in modifying the precipitation (D'Agostino et al., 2019; D'Agostino et al., 2020).

Climate models have long been useful tools to better understand past, present and future climate. Two decades ago, most LGM simulations were performed with stand-alone atmospheric or oceanic models with prescribed boundary conditions at the air-sea interface (Werner et al., 2001). To date it is still a challenge to reproduce the LGM climate although more advanced coupled models have been developed (Kageyama et al., 2017, 2021). The reason lies in the uncertainties associated with the

reconstructed LGM topography (Werner et al., 2018), as well as the large heterogeneity in the simulation results of various climate models (Kageyama et al., 2021). Spatial resolution is also an important factor affecting the modelled climate of the past (Shi and Lohmann, 2016; Shi et al., 2020; Lohmann et al., 2021).

Although cold and dry LGM climates have been captured in most modelling studies (Kageyama et al., 2021), in good agreement with the proxy record (Bartlein et al., 2011; Members et al., 2009), one question to be addressed here is what is the

relative contribution of individual factors in determining the anomalous LGM temperature and precipitation from its modern-day condition, such as greenhouse gases, ice sheets, as well as orbital parameters. The model results, based on NESM v1,

underscored the significant influence of greenhouse gases and ice sheets in shaping the anomalous LGM climates in contrast to the present day (Cao et al., 2019). Conducting additional sensitivity experiments with various models would facilitate model intercomparisons and provide a more comprehensive understanding of the impact of individual forcings and boundary conditions on the LGM climate. For this purpose, in the present study, we use a state-of-the-art climate model AWI-ESM with an unstructured ocean grid, to perform simulations under pre-industrial and LGM boundary conditions. We also conduct several sensitivity experiments to examine the role of individual or combined boundary conditions and forcings of the LGM.

## 2  Methodology

### 2.1  Model description

AWI-ESM is a state-of-the-art Earth system model developed at the Alfred Wegener Institute (Sidorenko et al., 2015; Rackow et al., 2018). The atmospheric component of AWI-ESM is represented by the general circulation model ECHAM6 (Stevens et al., 2013), which has been mainly developed at the Max-Planck Institute for Meteorology (MPI-M). The model also includes a Land-Surface Model (JSBACH) which is based on a tiling of the land surface and includes dynamic vegetation with 11 plant functional types and two types of bare surface (Loveland et al., 2000). The ice-ocean component is the Finite Element Sea Ice-Ocean Model (FESOM) formulated on unstructured meshes (Sidorenko et al., 2014). FESOM uses a multi-resolution approach, and therefore allows us to increase the resolution locally over the area of interest, while keeping the resolution elsewhere unchanged (Sidorenko et al., 2011). The coupled model AWI-ESM has been validated under modern climate conditions (Sidorenko et al., 2014, 2015; Rackow et al., 2018) and applied widely in simulating the climate of the past (Lohmann et al., 2020; Otto-Bliesner et al., 2021; Brierley et al., 2020; Renoult et al., 2020; Yang et al., 2022a), present (Shi and Lohmann, 2017) and future (Yang et al., 2020, 2022b).

### 2.2  Experimental design

The atmosphere grid applied in the present study is T63L47, which has a global mean spatial resolution of 1.875 x 1.875 degrees with 47 vertical levels. A spatially-variable resolution is used for the ice-ocean component (Fig. S1), ranging from about 100 km in the open ocean to 25 km over polar areas and 35 km for the equatorial belt and along coastlines. Vertically, there are 46 uneven layers in the ocean.

We first perform a background LGM and a pre-industrial (PI) simulation following the protocol of the Paleoclimate Modelling Intercomparison Project phase 4 (Kageyama et al., 2017). The land-sea mask, ice sheets and topography are prescribed according to the ICE6G reconstruction (Peltier et al., 2015). Orbital parameters are calculated based on the Berger equations (Berger, 1977). The initial condition for atmosphere and ocean in our PI experiment is derived from the Atmospheric Model Intercomparison Project (AMIP) and the World Ocean Atlas (WOA), respectively. We initiate our LGM simulation from a previous LGM run using ECHAM5-MPIOM (Werner et al., 2016). Both LGM and PI simulations are integrated for more than

**Table 1.** List of experiments and their boundary conditions/forcings.

| Name | Greenhouse gases | ice sheets | Orbital parameters |
|------|------------------|------------|--------------------|
| PI | 0k | 0k | 0k |
| LGM | 21k | 21k | 21k |
| LGM_G | 0k | 21k | 21k |
| LGM_I | 21k | 0k | 21k |
| LGM_O | 21k | 21k | 0k |
| LGM_GI | 0k | 0k | 21k |

1,300 model years with the simulated climate being in a quasi-equilibrium state for the final 100 years. The climatology of each climate variable is then represented by its average over the last 100 model years.

Four additional sensitivity experiments have been conducted according to the PMIP4 protocol (Kageyama et al., 2017), in which the land–sea mask stays the same as in the LGM experiment, but with different ice sheets, greenhouse gases, or Earth's orbital parameters prescribed (Table 1). In detail, the sensitivity experiments include:

(a) The LGM_G experiment, in which all boundary conditions and forcings are set to the LGM values except for the greenhouse gases, which are the same as for PI.

(b) The LGM_I experiment, in which all boundary conditions and forcings are set to the LGM values but the ice sheets are the same as for PI.

(c) The LGM_O experiment, in which all boundary conditions and forcings are set to the LGM values but the orbital parameters are the same as for PI. Note that this experiment is not proposed by PMIP4 (Kageyama et al., 2017) but is included in our experimental setup as it can offer an indication of the effect of a changed astronomical forcing.

(d) The LGM_GI experiment, in which all boundary conditions and forcings are set to the LGM values except for the greenhouse gases and ice sheets, which are the same as for PI.

We integrated each sensitivity simulation for at least 400 model years. The climatological patterns derived from the last 100 model years are used for the following analyses.

## 3 Result

### 3.1 Full-forced LGM

Our full-forced LGM experiment simulates a mean global cooling by -4 K in comparison to PI, which meets the estimate based on various proxy reconstruction (Annan and Hargreaves, 2013; Annan et al., 2022), but not consistent with the estimate of -6.5 to -5.7 K cooling from Tierney et al. (2020). With respect to PI, our LGM simulation yields a reduction in global precipitation by 6.5%. Thus the hydrological sensitivity of our model, defined as the slope between changes in global mean precipitation versus surface temperature, has the value of 1.625 %/K, which is relatively smaller as compared to other model studies. For

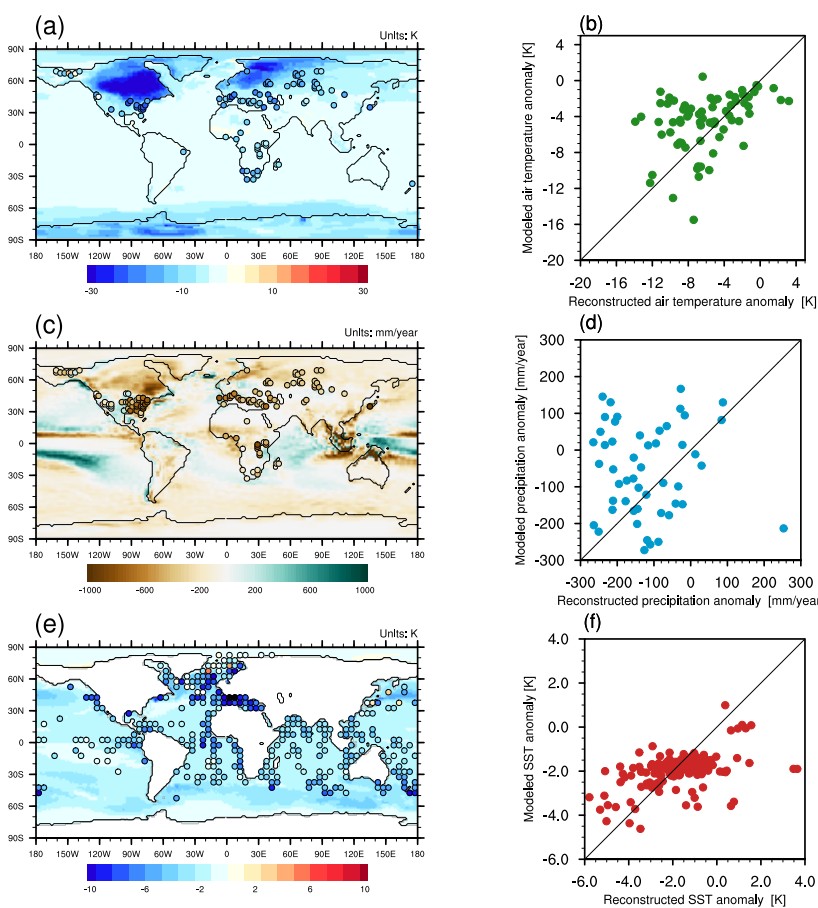

**Figure 1.** (a,c,e) Simulated (shading) and reconstructed (circles) anomalies between LGM and pre-industrial for annual mean (a) surface air temperature, (b) precipitation, and (c) sea surface temperature. Units are K, mm/year, and K, respectively. (b,d,f) Scatter plot of modeled versus reconstructed annual mean (b) surface air temperature, (d) precipitation, and (f) sea surface temperature.

example, Cao et al. (2019) yields a hydrological sensitivity of 2.2% K-1. According to Li et al. (2013), the values obtained from model ensemble mean are 2.64% /K and 2.28%/K for ocean and land, respectively. To assess the model's performance in representing spatial surface air temperature and precipitation changes in LGM versus PI over continents, we use here a set of pollen-based proxy records (Bartlein et al., 2011). Fig. 1a,b represents the simulated and reconstructed annual mean surface air temperature anomalies between the LGM and PI experiments. In response to the full LGM boundary conditions and

forcings, our simulated LGM surface air temperature is overall colder than for PI climate conditions, in good agreement with reconstructions. The most prominent characteristic is a pronounced cooling over the Laurentide and Fennoscandia ice sheets (up to -34 K). The simulated temperature anomalies over glacier-free areas range from 0 to -10 K. Our model shows good agreement between the simulated and observed temperature anomalies over the tropics, but tends to underestimate the cooling in some subtropical regions such as northern and southernmost parts of Africa, as well as the adjacent areas of the Northern

Hemisphere ice sheets. Additionally, two proxy records show warming over Alaska that may be due to the different land-sea distribution caused by the sea level drop between LGM and PI. In general, model and proxy agree on the dryness across most continents in the LGM versus PI (Fig. 1b). The most significant precipitation reduction occurs over Northern Hemisphere land-ice regions, North Africa, South Asia, as well as over the Tropical Warm Pool (TWP). Southern North America, the western Indian Ocean, and the southern edge of the Pacific rain-belt experience more precipitation in the LGM than in the PI.

In terms of LGM sea surface temperature (SST), we evaluate the performance of our model in comparison to reconstructed data derived from the multi-proxy approach for glacial ocean surface reconstruction (MARGO) (Members et al., 2009). In general, the simulated SST anomaly pattern mimics the observation (Fig. 1e), showing global cooling in the LGM compared to the present. The anomalies are relatively larger in the northern North Atlantic and the Pacific Oceans, as well as the Mediterranean Sea. The cooling across the tropical oceans are relatively weaker. In general, our model shows good agreement with the SST

reconstructions (Fig. 1e), but appears to underestimate the magnitude of the SST difference between LGM and PI (Fig. 1f). Some of the local warming over the Nordic Sea and Pacific Ocean, as indicated by proxy data, is not captured by our model.

In addition, the PMIP4 model ensemble also shows a global cooling which finds its largest expression over ice sheets and polar regions, as well as pronounced dryness over tropical monsoon domains and subpolar areas (Kageyama et al., 2021). Therefore our model agrees well with other climate models participating in the PMIP4.

## 3.2 Sensitivity experiments

In this section we examine the contributions of individual boundary conditions and forcings including the orbital parameters, greenhouse gases, and ice sheets to the simulated LGM climate.

a. Surface air temperature

As mentioned, our model yields a global mean surface temperature anomaly between LGM and PI of -4 K. Here, our

sensitivity experiments show that the LGM greenhouse gases, ice sheets, and orbital forcings respectively contribute to a global mean cooling of -2.05 K, -1.34 K and -0.03 K. The combination of LGM greenhouse gases and ice sheets lead to a global mean temperature anomaly of -3.32 K. The additional cooling of -0.68 K is likely contributed by the different land-sea distribution between LGM and PI, which is however not investigated in the present study (since the PI land-sea mask cannot be

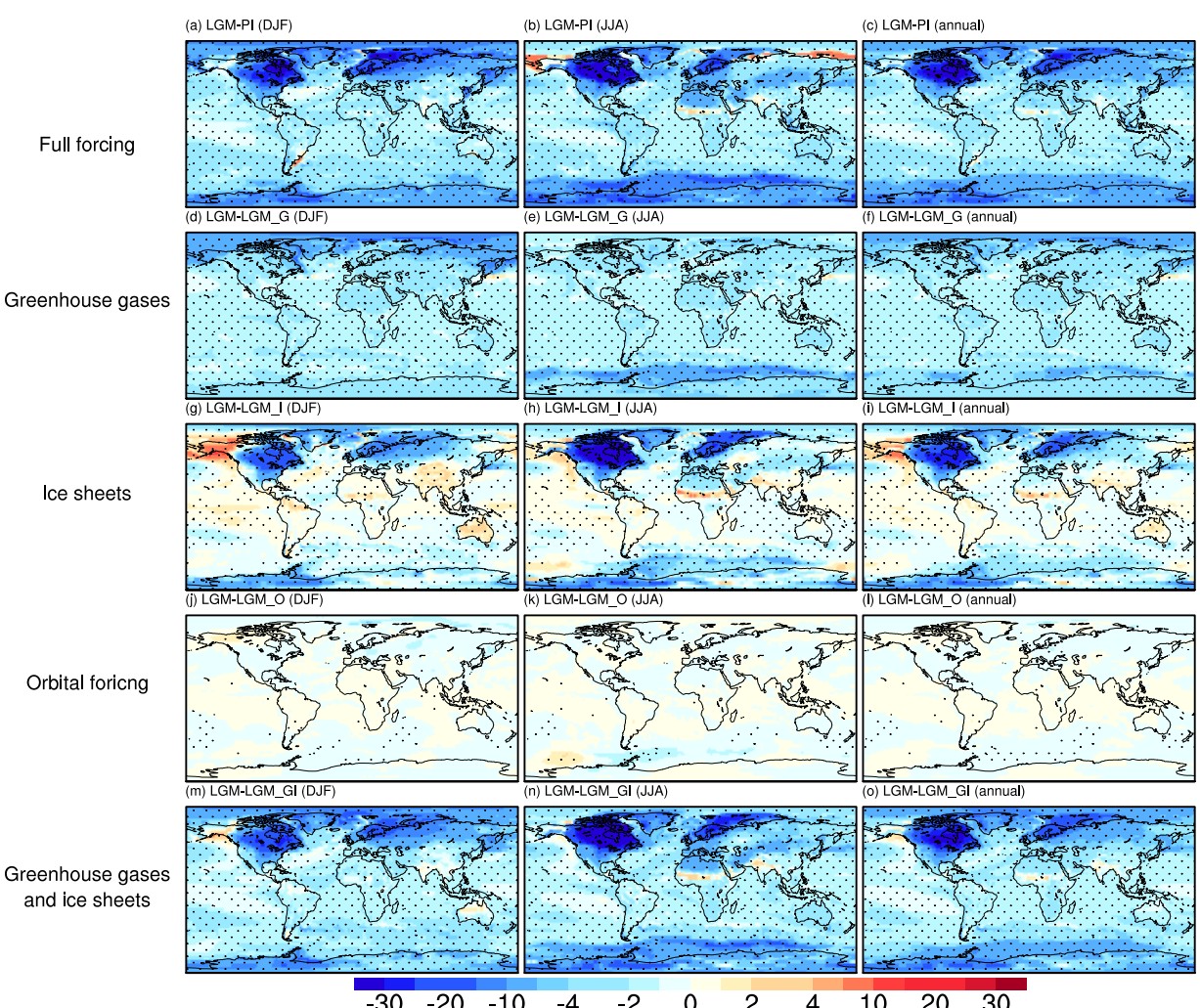

**Figure 2.** Surface air temperature anomalies for (a-c) LGM-PI, (d-f) LGM-LGM_G, (g-i) LGM-LGM_I, (j-l) LGM-LGM_O; and (m-o) LGM-LGM_GI. (a,d,g,j,m) are for DJF, (b,e,h,k,n) are for JJA, and (c,f,i,l,o) are for annual mean. The marked area has a significance level of greater than 95% based on Student's t-test. Units: K.

combined with the LGM ice sheet). This indicate the dominating effects of both greenhouse gas and ice sheets on determining

the global mean surface temperature change between LGM and PI, with the former playing a more important role.

  Spatially, the full-forced LGM simulates a general global cooling compared to PI, with the largest change occurring over the Laurentide and Scandinavian ice sheets, both in winter and in summer (Fig. 2a-c). Fig. 2d-f illustrate the differences in surface air temperature between LGM and LGM_G, highlighting the effect of greenhouse gases. As indicated in Fig. 2d-f, the reduction in greenhouse gases contributes to an overall cooling of the globe, with an amplified effect over the polar regions.

Comparing LGM and LGM_I, we find that the pronounced negative temperature anomalies over North America and northern Eurasia are driven by the presence of the Laurentide and Scandinavian ice sheets due to their high elevation and surface albedo (Fig. 2g-i). Moreover, localized cooling is observed over Greenland and Antarctica, led by an increase in land-ice extent and elevation. Another intriguing feature induced by the LGM ice sheets is a warming over most glacier-free continents of the Northern Hemisphere during boreal winter (Fig. 2g). The underlying mechanism for this warming lies in the intensification of

the Atlantic Meridional Overturning Circulation (AMOC) (Fig. S3c), which gives rise to a stronger poleward heat transport and thus to a warmer Northern Hemisphere. The effect of the LGM ice sheets (especially the topographical influence of the Laurentide ice sheet) on the AMOC is through the strengthening of westerly winds over the North Atlantic (Fig. S4). In particular, the North Atlantic subpolar gyre is reinforced by the strengthened zonal winds and tends to transport more saline seawater from low to high latitudes. As a consequence, salinity increases in the North Atlantic subpolar zone, resulting in a

less stratified ocean column and thus enhanced mixing at the main convection sites. Moreover, the cold air transported from the glaciers is advected towards the subpolar region of the North Atlantic, resulting in an increase in seawater density in the upper layers. This, in turn, serves to further enhance the strength of the AMOC. When comparing LGM and LGM_O we find no direct influence of the orbital parameters (Fig. 2j-l). Finally, Fig. 2m-o shows the temperature response to the combined forcings of LGM greenhouse gases and ice sheets, resulting in patterns similar to Fig. 2a-c, emphasizing the key role of greenhouses and

ice sheets in determining LGM surface air temperatures. As indicated above, our sensitivity experiments suggest that changes in greenhouse gases and ice sheets are the main drivers of the anomalous LGM surface temperature relative to PI. The solar insolation change does not have direct impact on LGM surface temperatures, but could shape the LGM climate via its long-term influences on the ice sheets. In addition, by comparing the global mean surface temperature between LGM and LGM_GI, the change in land-sea distribution might have a non-negligible impact.

Simulated sea ice anomalies are closely related to temperature changes. In general, in the LGM simulation we observe more sea ice coverage over both the Arctic and Southern oceans with respect to PI (Fig. 3a and Fig. 4a). The reduction in LGM greenhouse gases contributes to positive anomalies in sea ice concentration (Fig. 3b and Fig. 4b). The cooling effect of LGM ice sheets results in increased sea ice coverage over Baffin Bay and Nordic Sea (Fig. 3c). Reduced sea ice cover is evident in North Pacific when comparing LGM versus LGM_I (Fig. 3c), this is due to the regional warming as shown in Fig. 2g-i.

Moreover, the increased concentration of sea ice over the Southern Ocean indicated in Fig. 4c is due to enhanced northward heat transport associated with AMOC intensification, as well as hemispheric cooling facilitated by increased elevation of the Antarctic ice sheet. The combined effect of greenhouse gases and ice sheets forcings contributes to a pattern of sea ice concentration anomaly resembling that of the full-forced LGM simulation (Fig. 3e and Fig. 4e). However, for the Arctic

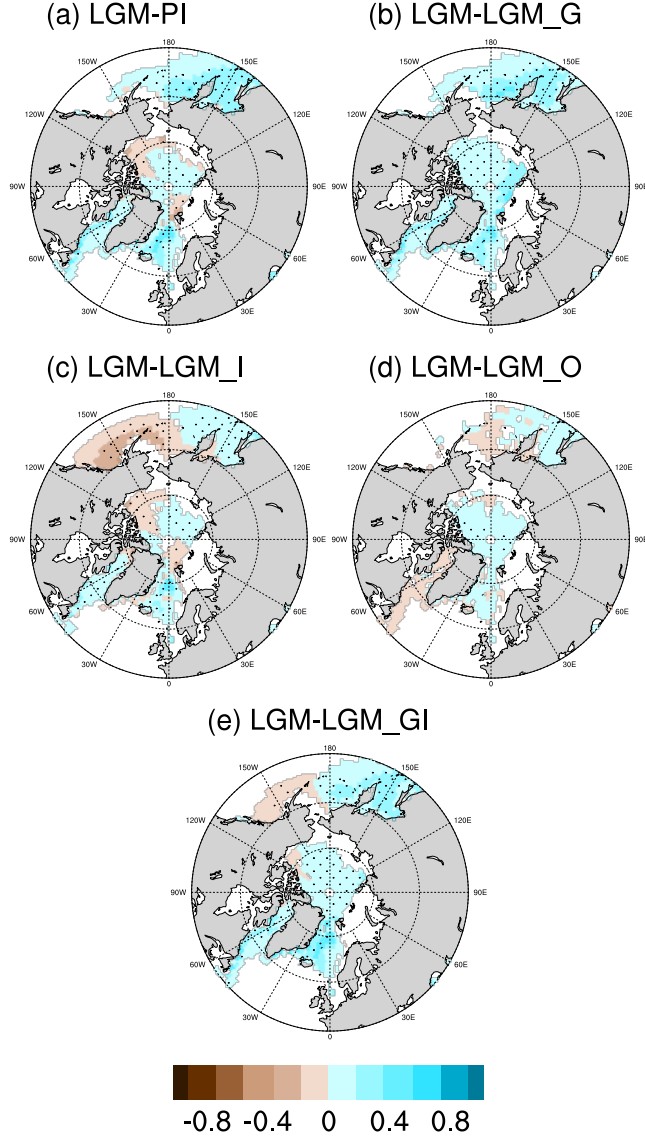

**Figure 3.** Anomalies in annual mean Arctic sea ice concentration for (a) LGM-PI, (b) LGM-LGM_G, (c) LGM-LGM_I, (d) LGM-LGM_O, and (e) LGM-LGM_GI. The marked area has a significance level of greater than 95% based on Student's t-test. Units are %.

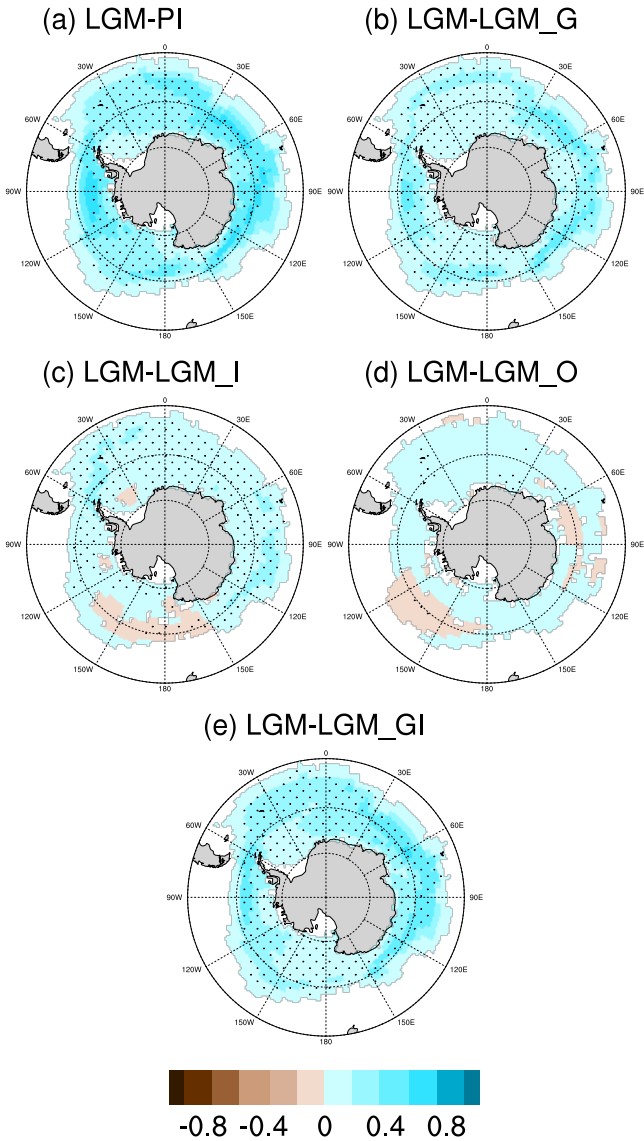

**Figure 4.** Anomalies in annual mean Southern Ocean sea ice concentration for (a) LGM-PI, (b) LGM-LGM_G, (c) LGM-LGM_I, (d) LGM-LGM_O, and (e) LGM-LGM_GI. The marked area has a significance level of greater than 95% based on Student's t-test. Units are %.

coastal areas, due to the different land-sea distribution, negative anomalies of sea ice concentration are found in the full-forced

LGM compared to PI. The decreased sea ice in North Pacific induced by LGM ice sheets is not evident in LGM and LGM_GI, indicating a non-linear response of sea ice to LGM forcings. Note that in our sensitivity experiments, the land-sea masks remain the same as in the LGM, therefore the differences in sea ice concentration do not reflect the influence of sea level changes.

    b. Precipitation

    Compared to PI, our full-forced LGM experiment represents a generally drier tropical climate in the boreal summer. Less

precipitation is simulated in LGM over the tropical rainbelt (Fig. 5b), which dominates the annual anomaly pattern (Fig. 5c). Drying over the Arctic, Southern Ocean, and glacier-covered regions can be observed throughout the year (Fig. 5a-c). The global mean precipitation as simulated in our full-forced LGM experiment is 2.62 mm/day, 0.19 mm/day lower than the PI condition. The mean precipitation anomaly between LGM and LGM_G (LGM_I) is -0.086 (-0.083) mm/day. The global mean precipitation change due to solar radiation forcing is only -0.0015 mm/month and is not statistically significant based on

Student's t-test. Finally, comparing LGM versus LGM_GI, we find that the LGM greenhouse gases and ice sheets together lead to a global drying by -0.181 mm/day. Our results suggest that both the LGM greenhouse gases and ice sheets are responsible for the drying conditions during the LGM. However, we find different spatial patterns of precipitation anomalies induced by the two forcings: Greenhouse gas forcing contributes to negative anomalies over TWP, Southern Ocean and northern Eurasia, but positive values over North Hemisphere monsoon regions i.e., tropical America, North Africa and parts of South Asia

(Fig. 5d-f). Ice sheet forcing causes dryness over most monsoon regions (i.e., North Africa, South Asia), plus northern part of North America and Australia, as well as wetness over southern part of North America (Fig. 5g-i), resembling the precipitation anomaly pattern between full-forced LGM and PI. Thus, our results suggest an important role played by the LGM ice sheets in deviating precipitation patterns. The dipole pattern of precipitation anomalies over North America is associated with reoriented winds led by the presence of the Laurentide ice sheet in the LGM. LGM ice sheets and greenhouse gases lead to drying and

wettening respectively, over North Africa and South Asia during boreal summer. However, the combination of the two forcings results in an even more pronounced dryness over North Africa and South Asia (Fig. 5n-o) than does the ice sheet forcing alone, indicating a nonlinear response of precipitation to multiple LGM boundary conditions and forcings.

    The general dryness due to the LGM ice sheets and greenhouse gases is caused by a weakening in the hydrological cycle: as the climate cools, the moisture of the air decreases. Fig. S5 indicates an overall decrease in low level specific humidity

when either LGM ice sheets or the LGM greenhouse gases are imposed, with the former leading to a much larger anomaly. One question arise here as to why the reduced greenhouse gases give rise to increased precipitation over Northern Hemisphere monsoon regions during boreal summer. To answer this question, we perform a moisture decomposition analysis between LGM and LGM_G for each Northern Hemisphere monsoon domain using the method described in D'Agostino et al. (2019). Following D'Agostino et al. (2019), the monsoon domain is defined as the continental area where the annual precipitation

range (i.e., the difference between summer and winter precipitation) exceeds 2 mm/day for the following regions:

    (1) North African (5–23.3°N, 20°W to 40°E).

    (2) South Asia (5–23.3°N, 70–120°E).

    (3) North American (5–30°N, 120–40°W).

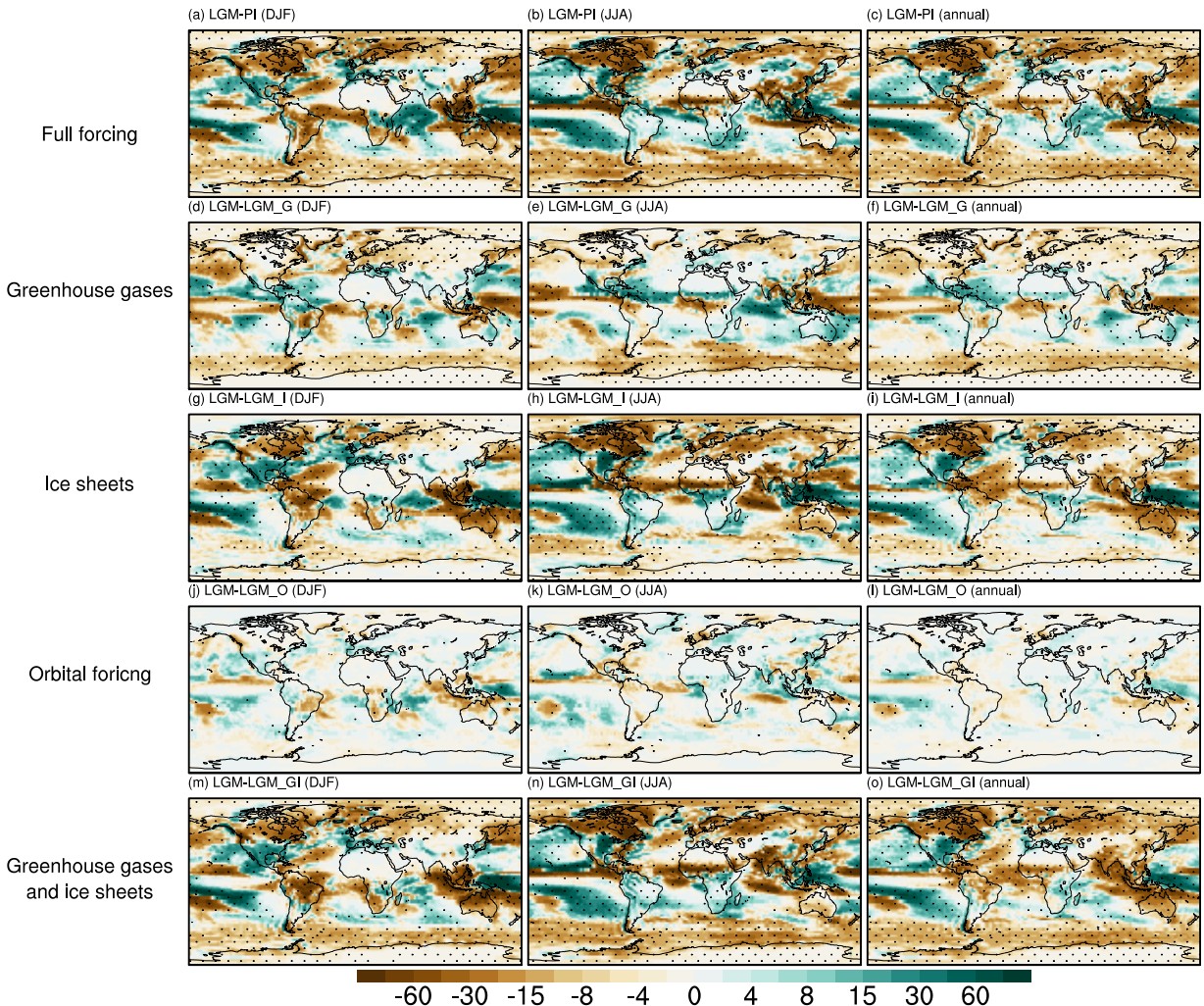

**Figure 5.** Precipitation anomalies for (a-c) LGM-PI, (d-f) LGM-LGM_G, (g-i) LGM-LGM_I, (j-l) LGM-LGM_O; and (m-o) LGM-LGM_GI. (a,d,g,j,m) are for DJF, (b,e,h,k,n) are for JJA, and (c,f,i,l,o) are for annual mean. The marked area has a significance level of greater than 95% based on Student's t-test. Units are mm/month.

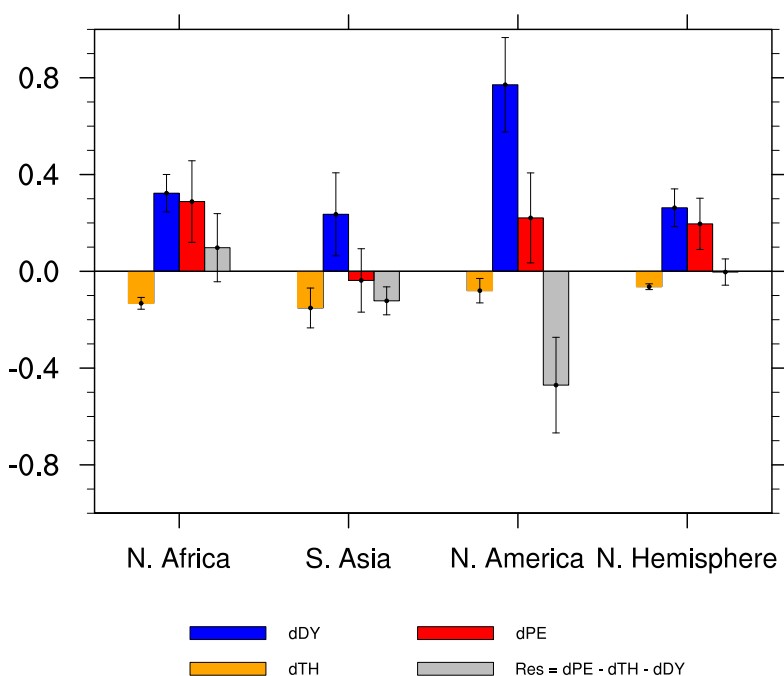

**Figure 6.** Regionally averaged changes in boreal summer net precipitation (dPE) and its thermodynamic (dTH), dynamic (dDY), and the residual (Res) components for LGM-LGM_G. Variances are represented by error bars. Units are mm/day.

Following D'Agostino et al. (2019), we decompose the anomalous moisture budget into thermodynamic (dTH) and dynamic (dDY) components as well as a residual (Res) term:

$$\rho_w g \delta(P - E) = - \int\limits_0^{p_s} \nabla(\delta \bar{\mathbf{q}} \bar{\mathbf{u}}) dp - \int\limits_0^{p_s} \nabla(\bar{\mathbf{q}} \delta \bar{\mathbf{u}}) dp - Res \tag{1}$$

where P, E, p, and q denotes precipitation, evaporation, pressure, and specific humidity respectively, u stands for the horizontal vector wind, and $\rho_w$ the water density. $\delta$ indicates the difference between the two simulations and overbars indicate monthly means. The first term on the right-hand side of Eq. (1) describes the contribution from the thermodynamic component (dTH) which is associated with differences in moisture convergence arising from specific humidity changes, and the second term represents the dynamic contribution (dDY) involving changes in mean atmospheric flow with unchanged moisture. The third term is the residual (Res) that mainly accounts for contributions from surface quantities and transient eddies, which are normally of minor importance. For more detailed information of the decomposition analysis we refer to (D'Agostino et al., 2019).

As shown in Fig. 6 and Fig. S6, the reduction of LGM greenhouse gases leads to an increase in net precipitation across all Northern Hemisphere monsoon regions. The only exception is the South Asia monsoon, which show no obvious tendency resulting from a partial compensation between the thermodynamic component (i.e. the one associated with humidity change) and the dynamic component (related to atmospheric mean flow). Surprisingly, most LGM precipitation-minus-evaporation (P-E) changes relative to LGM_G in the different monsoon domains are dynamically induced rather than thermodynamically constrained by the temperature drop (Fig. 6). In contrast the thermodynamic term contributes to a drying over most monsoon domains, only in North America does this component favor an increased net precipitation. This indicates that changes in the mean atmospheric flow (i.e., enhanced Hadley circulation) are responsible for the simulated monsoon strengthening. Moreover, over North America the residual term, associating with transient eddies and surface quantities, contributes to a dryness. In the next section we explore in more details the responses of the Hadley circulation to individual LGM boundary conditions and forcings.

c. Changes in the tropical atmospheric mean flow

Monsoon has been widely considered as a unique phenomenon in the Tropical Convergence Zone (ITCZ) where trade winds from the Northern and Southern Hemispheres converge, coinciding with the ascending branch of the Hadley circulation. Here, we investigate the Hadley circulation based on the meridional mass stream function calculated by Eq. (2) according to (D'Agostino et al., 2017), in which v denotes the meridional wind, g stands for the gravitational acceleration, a is the radius of the Earth, p and $\phi$ represent latitude and pressure respectively.

$$\psi_h = \frac{2\pi a cos\phi}{g} \int\limits_0^p v dp \tag{2}$$

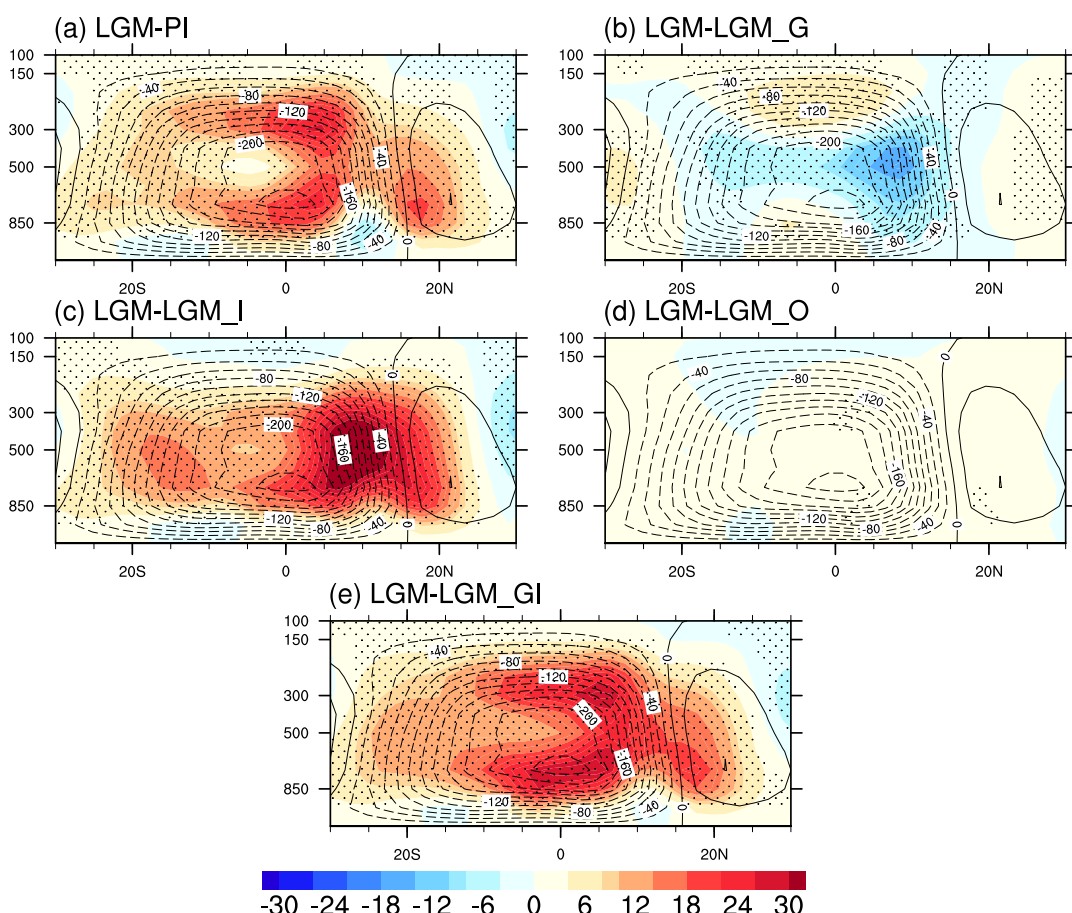

**Figure 7.** Simulated LGM JJAS zonal-mean meridional stream function (contours) and the anomalies of zonal-mean meridional stream function (shading) for (a) LGM minus PI, (b) LGM-LGM_G, (c) LGM-LGM_I, (d) LGM-LGM_O, and (e) LGM-LGM_GI. The marked area has a significance level of greater than 95% based on Student's t-test. Units are svp (1 svp = $10^9$ kg/s).

The anomalies of the JJAS-mean meridional mass stream function $\psi_h$ between LGM and other experiments are illustrated in Fig. 7. Relative to PI, our simulation with complete LGM boundary conditions and forcings suggests a weakening of the Southern Hemisphere Hadley cell which dominates the tropical circulation pattern during warm months (Fig. 7a). Such weakening is mainly due to the presence of LGM ice sheets (Fig. 7c). In contrast, the reduction of greenhouse gases reinforces the ascending branch of the Southern Hemisphere Hadley cell (Fig. 7b). This is consistent with previous studies indicating that increased $CO_2$ can result in larger gross moist stability in the tropics and thus a more stable atmosphere (Chou et al., 2013; D'Agostino et al., 2017). Overall, the effect of the LGM ice sheet forcing overwhelms that of the greenhouse gases and dominates the full-forced LGM tropical circulation (Fig. 7e).

Interhemispheric energy asymmetries, either in top-of-atmosphere radiation and/or in surface turbulent fluxes, alters the strength of the cross-equatorial Hadley circulation (see Hill et al. (2019) and literature therein). In the energy flux framework, for a given interhemispheric energy asymmetry, e.g. less energy in the Northern Hemisohere due to cold North Atlantic sea surface temperature and/or increased ice sheets, requires the ITCZ to shift equatorward, towards the more energetic hemisphere and a less intense southward cross-equatorial atmospheric heat transport (AHT_eq) operated by the Hadley cell (which strength is a proxy for AHT_eq).

In our simulations we find that the Southern Hemisphere Hadley cell is weaker in JJAS than PI because the atmospheric heat transport across the equator is also less intense. The full-forced LGM experiment has a reduced hemispheric energy contrast by 3.02 W/m$^2$ compared to the PI period, with LGM ice sheets being the dominant driver, contributing by 84% to the overall reduction. This decrease in hemispheric energy contrast favors a reduced cross-equatorial southward energy flux, leading to a equatorward contraction of the Hadley cells. Consequently, this induces a weakened (strengthened) Hadley circulation in the Southern (Northern) Hemisphere.

The energy flux framework has been widely used for explaining ITCZ shifts and Hadley circulation strength in response to a variety of forcing, e.g., AMOC weakening, changes in ice-sheet cover, orbital forcing, volcanic eruptions, aerosol distribution, (see Kang et al., 2008, 2009; Donohoe et al., 2013; Frierson et al., 2013; Marshall et al., 2014; McGee et al., 2014; Schneider et al., 2014; Boos and Korty, 2016; Jacobson et al., 2020; D'Agostino and Timmreck, 2022) in present climates and also at geological time scales in both paleoclimate records and simulations of the mid-Pliocene (Han et al., 2021), Younger Dryas ( 11 kyr BP), Last Glacial Maximum ( 21 kyr BP), Heinrich stadials (HS 1-4, e.g.,  18, 24, 31, 39 kyr BP) and some Dansgaard-Oeschger events (see Fig. 5 of Lynch-Stieglitz, 2017).

Furthermore, our results in terms of the Hadley cell changes can partly explain the precipitation anomalies. When the LGM ice sheets are imposed, the weakened mean atmospheric flow, together with the reduced specific humidity (Fig. S5 ) caused by the temperature drop, contribute to a drying condition, whereas reduced greenhouses gases tend to reinforce the mean tropical circulation and dynamically enhance the summer monsoon precipitation in the Northern Hemisphere.

Following D'Agostino et al. (2020), we compute also the Pacific Walker circulation (PWC), we analyze the zonal mass stream function ($\psi_w$) — calculated using divergent component of the wind ($u_D$) as in Eq. (3) — integrated over equatorial Pacific region (10°S-10°N, 110°E-90°W):

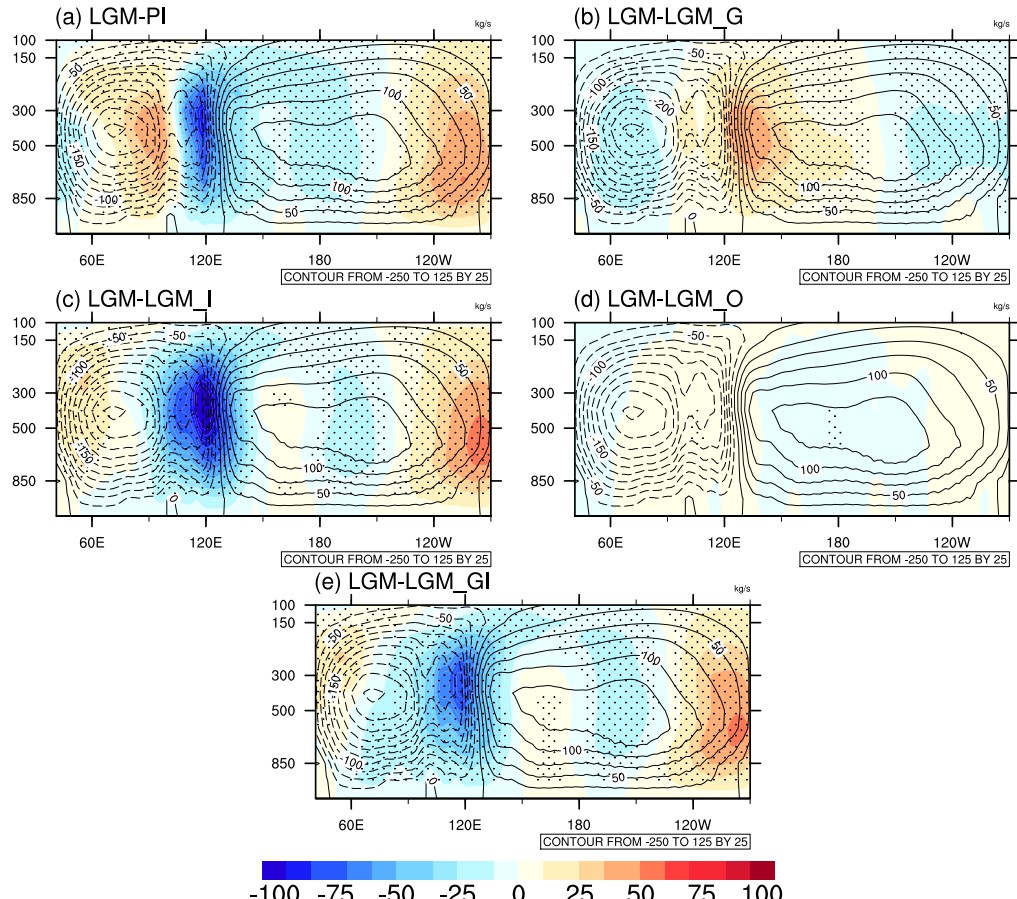

**Figure 8.** Simulated LGM annual mean zonal mass stream function (contours) and its anomaly (shading) for (a) LGM-PI, (b) LGM-LGM_G, (c) LGM-LGM_I, (d) LGM-LGM_O, and (e) LGM-LGM_GI. The marked area has a significance level of greater than 95% based on Student's t-test. Units are svp (1 svp = $10^9$ kg/s.).

$$\psi_w = 2\pi a \int\limits_0^p u_D \frac{dp}{g} \tag{3}$$

The large-scale pattern of the annual mean zonal mass stream function $\psi_w$ along the equator in the LGM is characterized by an overall weakening and eastward shift of the PWC with regards to PI: negative anomalies are found in Fig. 8a for the western part of PWC between 110°E and 150°W, with the most prominent changes occurring around the ascending branch,

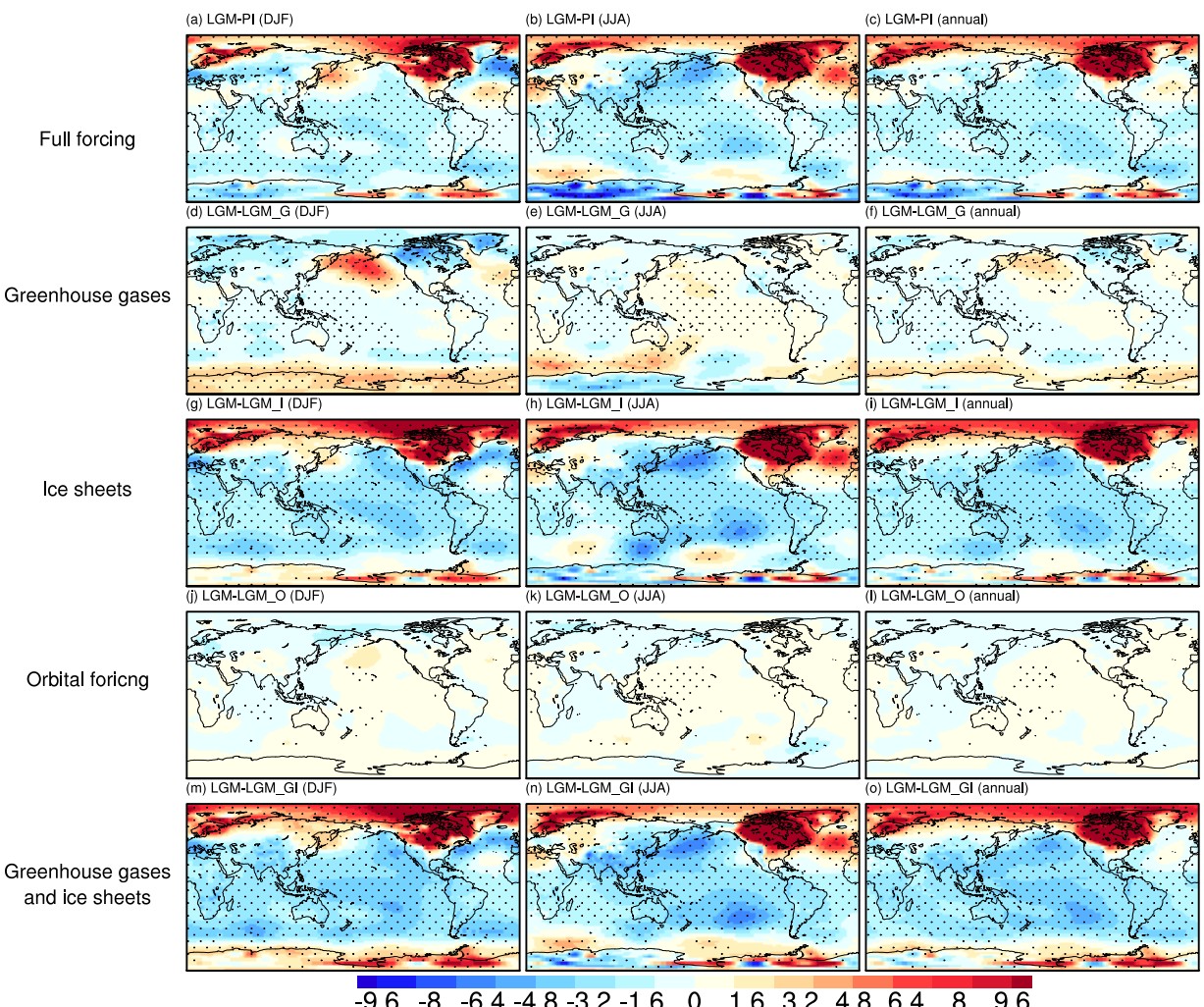

**Figure 9.** Sea leval pressure anomalies for (a-c) LGM-PI, (d-f) LGM-LGM_G, (g-i) LGM-LGM_I, (j-l) LGM-LGM_O; and (m-o) LGM-LGM_GI. (a,d,g,j,m) are for DJF, (b,e,h,k,n) are for JJA, and (c,f,i,l,o) are for annual mean. The marked area has a significance level of greater than 95% based on Student's t-test.The marked area has a significance level of greater than 95% based on Student's t-test. Units: hPa.

while positive anomalies can be observed east of 150°W. Such pattern of anomaly is in agreement with most PMIP models (Tian and Jiang, 2020). Similar results regarding the response of the PWC to LGM ice sheet forcing are found in Fig. 8c. In contrast, the reduction of LGM greenhouse gases has the opposite effect (Fig. 7b), which can, however, be counteracted by the topographical influence (Fig. 7e). Moreover, the effect of the astronomical forcings on PWC is negligible (Fig. 7d). The PWC changes are closely related to changes in sea level pressures (SLP). As shown in Fig. 9, the forcing of the LGM ice sheets (greenhouse gases) results in negative (positive) SLP anomalies over North Pacific, and thus a weakened (strengthened) low-level trade winds over the equatorial Pacific. In addition, previous studies have shown the relationship between PWC intensity and summer monsoon in South Asia and North Africa (Tian et al., 2018; Chang and Li, 2000). More specifically, increased summer monsoon precipitation favors enhanced convective heating in the monsoon regions, which induces anomalous easterly winds at the equatorial plane over western Pacific through a Kelvin wave response. Thus, the decrease (increase) in summer monsoon intensity is also responsible for the weakened (strengthened) PWC when the LGM ice sheets (greenhouse gases) are imposed.

d. El Niño-Southern Oscillation (ENSO) variability

El Niño-Southern Oscillation (ENSO) is a major source of global interannual variability. In this section, we aim to quantify the response of the ENSO variability to individual LGM boundary conditions and forcings. The ENSO variability is represented by the standard deviation of monthly sea surface temperature (SST) while removing seasonal variance. Our analysis reveals that the ENSO variability is substantially stronger in the full-forced LGM than in the PI (Fig. 10a,b). Even after removing the effects of LGM greenhouse gases or orbital parameters, the magnitude of ENSO remains high (Fig. 10c,e). This suggests minor impacts from the greenhouse gases and orbital parameters in driving ENSO variability during the LGM. Furthermore, we find that the ENSO variability in both flat-ice experiments, i.e. LGM_I and LGM_GI, is comparable to that in the PI experiment, indicating a predominating influence from the LGM continental ice sheets. Previous model studies using CSM1.4 and ECBilt-CLIO concur with our finding that ENSO was more intense in the LGM than at present (Otto-Bliesner et al., 2003; An et al., 2004; Peltier and Solheim, 2004). However, a later simulation using CCSM (version 3) showed the opposite result (Otto-Bliesner et al., 2006), demonstrating a model-dependency of ENSO behavior under LGM climate background, a point that was further confirmed by intercomparison of multiple LGM simulations (Zheng et al., 2008). In a more recent study, foraminifera-based LGM temperature variability for equatorial Pacific sites implied a weakened ENSO during the LGM relative to mid-to-late Holocene (Ford et al., 2015). However, the oxygen isotopic ratios of individual foraminifera obtained from deep-sea sediments provide evidence of increased LGM ENSO activity (Koutavas and Joanides, 2012). The large heterogeneity in LGM ENSO intensity across different studies emphasizes the importance of considering multiple factors when examining ENSO variability and its determinants. Further research is needed to fully understand the complex interactions between various factors and their impact on ENSO variability.

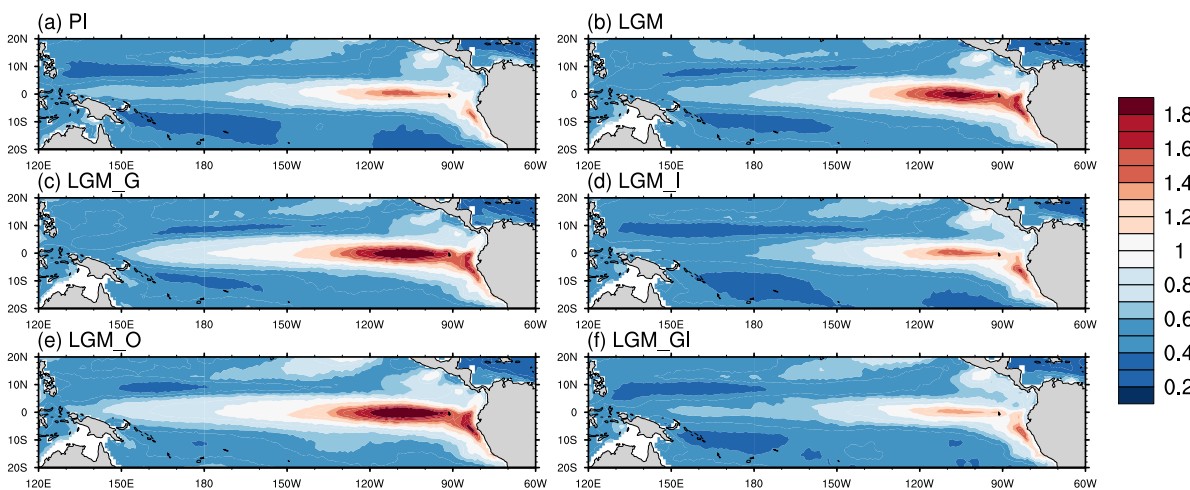

**Figure 10.** Standard deviation of monthly sea surface temperature for the tropical Pacific Ocean, with seasonal variance being removed. Units are K.

## 4  Discussion and conclusions

As the most recent glacial interval, the LGM has been examined in many previous studies with numerous models (Sime et al., 2016; Kageyama et al., 2021). Based on both observational reconstructions and model results, the LGM had a colder and dryer climate relative to present (Bartlein et al., 2011; Members et al., 2009). However, less is known about the individual contribution of LGM boundary conditions and external forcings such as orbital parameters, greenhouse gases and ice sheets, which have only been examined by a few studies to date. An example is the lowered-$CO_2$ experiment, which pointed to the important role of reduced $CO_2$ on global cooling (Brady et al., 2013). The relative role of oceanic heat transport and orography on the LGM climate was examined by using an atmospheric general circulation model of low resolution and intermediate complexity (Romanova et al., 2006) , their results indicated a strong dependence of the hydrological cycle and Northern Hemisphere atmospheric circulation on the orography and different oceanic heat transports. Similarly, some other studies also revealed that the North Atlantic westerlies during the LGM can be largely affected by the Laurentide ice sheet (Cook and Held, 1988; Marsiat and Valdes, 2001; Kageyama and Valdes, 2000), with a profound impact on Atlantic storm-tracks and northern Europe snowfall (Kageyama and Valdes, 2000). The LGM Northern Atmosphere circulation pattern can also be shaped by different prescribed sea surface temperatures, leading to regional climate changes during boreal winter (Marsiat and Valdes, 2001). In a more recent study, multiple sensitivity experiments using the coupled climate model NESM-v1 were performed to assess the relative influence of different external forcings (Cao et al., 2019), which suggested that the LGM cooling and dryness is mainly caused by changes in greenhouse gases and ice sheets, while solar radiation has only a minor impact.

In the present study we quantify the relative importance of Earth's orbit, as well as other impacting elements such as greenhouse gases and ice sheets, in determining the LGM climate. For this purpose, we perform a series of LGM experiments using a state-of-the-art Earth system model AWI-ESM, in which different combinations of boundary conditions and forcings have been applied in accordance with the protocols of PMIP4 (Kageyama et al., 2017). In good agreement with proxy records (Bartlein et al., 2011; Members et al., 2009) as well as other PMIP4 models (Kageyama et al., 2021), a generally colder and drier climate is simulated in our full-forced LGM experiment compared to the present. Our sensitivity experiments indicate that the LGM ice sheets and greenhouse gases together determine the general cooling and drying of the LGM in comparison to the PI, but with regional differences. The reduction of greenhouse gases in the LGM leads to non-uniform global cooling with a polar amplification effect. The presence of LGM ice sheets, primarily the Laurentide and the Scandinavian ice sheets, leads to local cooling due to their high surface albedo and elevation. We find substantial drying of the Northern Hemisphere monsoon regions, resulting from a compensation between the opposite influences of LGM ice sheets and greenhouse gases. A drying related to LGM ice sheets is associated with weakened atmospheric circulation and reduced moisture caused by the temperature drop. An analysis of LGM moisure budget changes indicates that enhanced atmospheric mean flow is responsible for the wettening tendency when LGM greenhouse gases are imposed. Combining the two forcings, the effect of the ice sheets dominates over that of the GHG and drives an overall decrease in LGM precipitation. Additionally, we observe that both the LGM ice sheets and the reduction in greenhouse gases contribute to the intensification of the AMOC (Fig. S3b,c,e). This intensification occurs through the strengthening of the North Atlantic westerlies and the increase in surface water density, respectively. In contrast,

changes in orbital parameters have minimal impact on the meridional circulation of the ocean (Fig.S3d). However, in the full-forcing experiment, the enhancement of the AMOC is not as pronounced as that caused by greenhouse gas forcing alone (Fig. S3b), ice sheet forcing alone (Fig. S3c), or the combination of the two (Fig. S3e). This phenomenon suggests either a non-linear response of the AMOC to individual LGM boundary conditions and forcings or a significant influence exerted by 350 the changes in land-sea distribution between LGM and PI.

Simulations of the Representative Concentration Pathway (RCP) global warming scenario indicated a positive correlation between global warming and monsoon rainfall (Acosta Navarro et al., 2017). But it should be noted that the monsoon precipitation change under global warming is not spatially uniform. For example, under future warming scenarios, South Asian summer monsoon was found to be weakened but the East Asian summer monsoon and the North Africa summer monsoon strengthened 355 (Li et al., 2022; Han et al., 2023). Previous studies have shown that increased $CO_2$ gives rise to enhanced moisture flux convergence resulting from increased atmospheric moisture and enhanced surface evaporation (Hsu et al., 2013; Lee and Wang, 2014), helping to increase monsoon precipitation. On the other hand, the $CO_2$ rise weakens the atmospheric circulation, which is caused by increased stability in the tropical region where lapse rate of temperature follows moist adiabats (Held and Soden, 2006). The weakened atmospheric circulation tends to dynamically reduce the tropical rainfall. Therefore the change in re- 360 gional monsoon precipitation in a warmer climate results from the compensation between the effects from enhanced moisture flux convergence and weakened atmospheric circulation (D'Agostino et al., 2019; Endo and Kitoh, 2014). Our model results, though based on a LGM background state, show that the decrease in greenhouse gases projects wettening of the Northern Hemisphere monsoon regions. Our analysis on moisture budget decomposition shows that reduction of greenhouse gases leads to a positive contribution from the dynamic component (i.e., strengthening of mean atmospheric circulation) to net precipitation 365 over monsoon domains, which cancels the negative contribution from reduced specific humidity (the thermodynamic term). Thus, our results indicate a potential climate-dependency of the Northern Hemisphere summer monsoon response to changes in greenhouse gases.

In agreement with previous studies (Romanova et al., 2006; Cook and Held, 1988; Kageyama and Valdes, 2000), the large scale atmospheric circulation of the LGM as simulated in the present study is largely shaped by the presence of the LGM ice 370 sheets. Although the direct impact of orbital forcings on LGM temperature, precipitation, ENSO, and atmospheric circulation is considered negligible due to the small change in precession between the LGM and PI periods, it is important to acknowledge that Earth's orbital changes primarily influence the glacial climate indirectly through internal climatic feedback processes (e.g., via its long-term influences on the ice sheets and greenhouse gases).

*Data availability.* The raw model outputs presented in this study can be accessed from https://doi.org/10.5281/zenodo.8063332.

375 *Author contributions.* X. Shi conceived the idea and designed the experiments. All authors contributed equally to the writing of the manuscript.

*Competing interests.* The authors have declared that no competing interests exist.

*Acknowledgements.* The present study is supported by the National Natural Science Foundation of China (NSFC) (grant no. 42206256) and the German Federal Ministry of Education and Science (BMBF) PalMod II WP 3.3 (grant no. 01LP1924B); Roberta D'Agostino is funded by the Deutsche Forschungsgemeinschaft (DFG, German Research Foundation) under Germany's Excellence Strategy – EXC 2037 Climate,
380 Climatic Change, and Society (CLICCS) - Cluster of Excellence Hamburg, A4 African and Asian Monsoon Margins – Project Number: 390683824 The AWI-ESM simulations were conducted on Deutsche Klimarechenzentrum (DKRZ) and AWI supercomputer (Ollie).

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
