# Peer review of "Unraveling the complexities of Last Glacial Maximum climate: the role of individual boundary conditions and forcings"

_Climate of the Past, 2023_

## Referee Comment (RC1)

This paper evaluates quantify the relative importance of individual boundary conditions and forcings, including greenhouse gases, ice sheets, and Earth's orbital parameters, on determining Last Glacial Maximum (LGM) climate using AWI-ESM. The results reveal that both the greenhouse gases and ice sheets play a major role on defining the anomalous LGM surface temperature compared to today. And the Northern Hemisphere monsoon precipitation is influenced by the opposing effects of LGM greenhouse gases and ice sheets. These large-scale changes are related to the AMOC and sea ice-albedo feedbacks, and hence modulate the atmospheric dynamics. I think that this paper shows an interesting topic, thus, I recommend the publication of this manuscript. Before publication, some minor issues that need to be addressed.

**General comments:**
1. This paper shows that the the enhanced AMOC exist in Ice sheet simulations, causing temperature anomalies over Northern Hemisphere. So, how about the AMOC changes induced by the CO2 changes?
2. An interesting result is that the weakening of the Southern Hemisphere Hadley cell in Figure 7, but what is the reason of this change? Maybe need further discussion of this. Besides, some studies, such as Han et al. (2022), consider the influence of imbalance of atmospheric energy budget on Hadley cell and ITCZ, and the Hadley cell change is because of the imbalance of atmospheric energy budget?

*Han, Z., Zhang, Q., Li, Q., Feng, R., Haywood, A.M., Tindall, J.C., Hunter, S.J., Otto-Bliesner, B.L., Brady, E.C., Rosenbloom, N. and Zhang, Z., 2021. Evaluating the large-scale hydrological cycle response within the Pliocene Model Intercomparison Project Phase 2 (PlioMIP2) ensemble. Climate of the Past, 17(6), pp.2537-2558.*

3. How about the standard deviation of each terms in Figure 6. Suggest to add the error bar in each term in Figure 6.
4. How about the spatial distribution of the moisture budget terms? Suggest to plot this figure in Supplementary Materials.
5. Which season in Figures 3, 8, 9, 10? Need to clarify.
6. The nonuniform warming pattern under global warming can influence the monsoon precipitation as well. Thus, the summer monsoon is not consistently weakened in the future warming scenarios. For example, Li et al. (2022) indicate that the South Asian summer monsoon is weakened but the East Asian summer monsoon is enhanced in the future warming scenarios. Han et al. (2022) show the North African summer monsoon is strengthened under SSP5-8.5 scenarios. Thus, I think the following sentences may need to be organized: "Therefore the increased monsoon precipitation in a warmer climate results from the compensation between the effects from enhanced moisture flux convergence and weakened atmospheric circulation".

*Li, T., Wang, Y., Wang, B., Ting, M., Ding, Y., Sun, Y., He, C. and Yang, G., 2022. Distinctive South and East Asian monsoon circulation responses to global warming. Science Bulletin, 67(7), pp.762-770.*

*Han, Z., Li, G. and Zhang, Q., 2022. Changes in Sahel summer rainfall in a global warming climate: contrasting the mid-Pliocene and future regional hydrological cycles. Climate Dynamics, pp.1-18.*

7. Please test the significance of variable changes in Figure 7 and 8.

**Specific comments:**
1. The coloarbar and labels n Figure 10 need to be larger. And the labels of latitude and longitude is too small as well.
2. "Modeled" suggest to change to "simulated" in line167.
3. Please check the whole paper, sometimes there is "ice sheet", but sometimes is "ice sheets".

---

## Author Comment (AC2)

**Response letter**

Xiaoxu Shi[1,2], Martin Werner[2], Hu Yang[1,2], Roberta D'Agostino[3,4], Jiping Liu[5,1], Chaoyuan Yang[1], and Gerrit Lohmann[2]

[1]Southern Marine Science and Engineering Guangdong Laboratory (Zhuhai), Zhuhai, China
[2]Alfred Wegener Institute, Helmholtz Center for Polar and Marine Research, Bremerhaven, Germany
[3]Max-Planck-Institute for Meteorology, Hamburg, Germany
[4]Department of Civil, Environmental and Mechanical Engineering, University of Trento, Trento, Italy
[5]School of Atmospheric Sciences, Sun Yat-sen University, Zhuhai, China

**Dear Reviewers,**

**Thank you very much for your positive and constructive comments. In the following, we present our point-to-point responses. Our answers to your comments are written in blue.**

**Thanks again for your time and efforts.**

5    **Best,**

**Xiaoxu**

**1    Comments from Reviewer 1**

This paper evaluates quantify the relative importance of individual boundaryconditions and forcings, including greenhouse gases, ice sheets, and Earth's orbital parameters, on determining Last Glacial Maximum (LGM) climate using AWI-ESM.

10    The results reveal that both the greenhouse gases and ice sheets play a major role ondefining the anomalous LGM surface temperature compared to today. And the Northern Hemisphere monsoon precipitation is influenced by the opposing effects of LGM greenhouse gases and ice sheets. These large-scale changes are related to the AMOC and sea ice-albedo feedbacks, and hence modulate the atmospheric dynamics. I think that this paper shows an interesting topic, thus, I recommend the publicationof this manuscript. Before publication, some minor issues that need to be addressed.

15    **We really appreciate your positive comments and suggestions, which have significantly improved our manuscript. We have modified our paper accordingly. Details are as follows:**

General comments.

1. This paper shows that the the enhanced AMOC exist in Ice sheet simulations, causing temperature anomalies over Northern Hemisphere. So, how about the AMOC changes induced by the CO2 changes?

20    **Thanks for the comment, we now added the change of AMOC for all sensitivity experiments in supplementary Fig. S3. We also discussed in the revised manuscript, as following (please also refer to line 343-350 in the revised manuscript):**

*"Additionally, we observe that both the LGM ice sheets and the reduction in greenhouse gases contribute to the intensification of the AMOC (Fig. S3b,c,e). This intensification occurs through the strengthening of the North Atlantic westerlies and the increase in surface water density, respectively. In contrast, changes in orbital parameters have minimal impact on the meridional circulation of the ocean (Fig. S3d). However, in the full-forcing experiment, the enhancement of the AMOC is not as pronounced as that caused by greenhouse gas forcing alone (Fig. S3b), ice sheet forcing alone (Fig. S3c), or the combination of the two (Fig. S3e). This phenomenon suggests either a non-linear response of the AMOC to individual LGM boundary conditions and forcings or a significant influence exerted by the changes in land-sea distribution between LGM and PI."*

2. An interesting result is that the weakening of the Southern Hemisphere Hadley cell in Figure 7, but what is the reason of this change? Maybe need further discussion of this. Besides, some studies, such as Han et al. (2022), consider the influence of imbalance of atmospheric energy budget on Hadley cell and ITCZ, and the Hadleycell change is because of the imbalance of atmospheric energy budget?

Han, Z., Zhang, Q., Li, Q., Feng, R., Haywood, A.M., Tindall, J.C., Hunter, S.J., Otto-Bliesner, B.L., Brady, E.C., Rosenbloom, N. and Zhang, Z., 2021. Evaluating the large-scale hydrological cycle response within the Pliocene Model Intercomparison Project Phase 2 (PlioMIP2) ensemble. Climate of the Past, 17(6), pp.2537-2558.

**Thanks so much for this constructive comment. According to the reviewer, we calculated the change in the hemispheric asymmetry of the energy flux which is found to be responsible for the change in Hadley circulation. In the revised paper, we modified the related texts into (please also refer to line 251-269 in the revised manuscript):**

*"Interhemispheric energy asymmetries, either in top-of-atmosphere radiation and/or in surface turbulent fluxes, alters the strength of the cross-equatorial Hadley circulation (see Hill et al. (2019) and literature therein). In the energy flux framework, for a given interhemispheric energy asymmetry, e.g. less energy in the Northern Hemisohere due to cold North Atlantic sea surface temperature and/or increased ice sheets, requires the ITCZ to shift equatorward, towards the more energetic hemisphere and a less intense southward cross-equatorial atmospheric heat transport (AHT_eq) operated by the Hadley cell (which strength is a proxy for AHT_eq)."*

*"In our simulations we find that the Southern Hemisphere Hadley cell is weaker in JJAS than PI because the atmospheric heat transport across the equator is also less intense. The full-forced LGM experiment has a reduced hemispheric energy contrast by 3.02 $W/m^2$ compared to the PI period, with LGM ice sheets being the dominant driver, contributing by 84% to the overall reduction. This decrease in hemispheric energy contrast favors a reduced cross-equatorial southward energy flux, leading to a equatorward contraction of the Hadley cells. Consequently, this induces a weakened (strengthened) Hadley circulation in the Southern (Northern) Hemisphere."*

*"The energy flux framework has been widely used for explaining ITCZ shifts and Hadley circulation strength in response to a variety of forcing, e.g., AMOC weakening, changes in ice-sheet cover, orbital forcing, volcanic eruptions, aerosol distribution, (see Kang et al., 2008, 2009; Donohoe et al., 2013; Frierson et al., 2013; Marshall et al., 2014; McGee et al., 2014; Schneider et al., 2014; Boos and Korty, 2016; Jacobson et al., 2020; D'Agostino and Timmreck, 2022) in present*

*climates and also at geological time scales in both paleoclimate records and simulations of the mid-Pliocene (Han et al., 2021), Younger Dryas ( 11 kyr BP), Last Glacial Maximum ( 21 kyr BP), Heinrich stadials (HS 1-4, e.g., 18, 24, 31, 39 kyr BP) and some Dansgaard-Oeschger events (see Fig. 5 of Lynch-Stieglitz, 2017)."*

3. How about the standard deviation of each terms in Figure 6. Suggest to add the error bar in each term in Figure 6.

**Thanks for the suggestion, we now added error bars in Fig. 6 in the revised manuscript (at page 13), and we also updated the caption accordingly.**

4. How about the spatial distribution of the moisture budget terms? Suggest to plot this figure in Supplementary Materials.

**According to the comment of the reviewer, we now added Supplementary Fig. S6 in the revised paper for describing the spatial distribution of the moisture budget terms.**

5. Which season in Figures 3, 8, 9, 10? Need to clarify.

**Thanks for the comment, in the revised version we gave more detailed information in the captions of all mentioned figures.**

6. The nonuniform warming pattern under global warming can influence the monsoon precipitation as well. Thus, the summer monsoon is not consistently weakened in the future warming scenarios. For example, Li et al. (2022) indicate that the South Asian summer monsoon is weakened but the East Asian summer monsoonis enhanced in the future warming scenarios. Han et al. (2022) show the North African summer monsoon is strengthened under SSP5-8.5 scenarios. Thus, I think the following sentences may need to be organized: "Therefore the increased monsoon precipitation in a warmer climate results from the compensation between the effects from enhanced moisture flux convergence and weakened atmospheric circulation".

Li, T., Wang, Y., Wang, B., Ting, M., Ding, Y., Sun, Y., He, C. and Yang, G., 2022. Distinctive South and East Asian monsoon circulation responses to global warming. Science Bulletin, 67(7), pp.762-770.

Han, Z., Li, G. and Zhang, Q., 2022. Changes in Sahel summer rainfall in a global warming climate: contrasting the mid-Pliocene and future regional hydrological cycles. Climate Dynamics, pp.1-18.

**We agree that the monsoon precipitation change under global warming is region-dependent. In the revised paper, we modified the related texts into (please also refer to line 351-361 in the revised manuscript):**

*"Simulations of the Representative Concentration Pathway (RCP) global warming scenario indicated a positive correlation between global warming and monsoon rainfall (Acosta Navarro et al., 2017). But it should be noted that the monsoon precipitation change under global warming is not spatially uniform. For example, under future warming scenarios, South Asian summer monsoon was found to be weakened but the East Asian summer monsoon and the North Africa summer monsoon strengthened (Li et al., 2022; Han et al., 2023). Previous studies have shown that increased $CO_2$ gives rise to enhanced moisture flux convergence resulting from increased atmospheric moisture and enhanced surface evaporation (Hsu et al., 2013; Lee and Wang, 2014), helping to increase monsoon precipitation. On the other hand, the $CO_2$ rise weakens the atmospheric circulation, which is caused by increased stability in the tropical region where lapse rate of temperature follows moist adiabats (Held and Soden, 2006). The weakened atmospheric circulation tends to dynamically reduce the*

*tropical rainfall. Therefore the change in regional monsoon precipitation in a warmer climate results from the compen-*
*sation between the effects from enhanced moisture flux convergence and weakened atmospheric circulation (D'Agostino et al., 2019; Endo and Kitoh, 2014). "*

7. Please test the significance of variable changes in Figure 7 and 8.

**Thanks for the suggestion, in the revised manuscript we have added the significance information in both Fig. 7 and Fig. 8 (at page 15 and 17).**

Specific comments:

1. The coloarbar and labels n Figure 10 need to be larger. And the labels of latitude and longitude is too small as well.

**Thanks, we have updated Fig. 10 in the updated version according to the comment (at page 19).**

2. "Modeled" suggest to change to "simulated" in line167.

**Thanks, we have changed "Modeled" to "Simulated" in that sentence. Moreover, we also made the same changes for other parts of the paper.**

3. Please check the whole paper, sometimes there is "ice sheet", but sometimes is "ice sheets".

**We have made corrections according to the reviewer's comment. We stick to "ice sheets" in the revised paper, but "ice sheet" is used when a specific ice sheet is mentioned, for example "the Laurentide ice sheet", it is also used in the term "ice sheet forcing".**

**2 Comments from Reviewer 2**

Xiaoxu Shi and co-authors presented a comprehensive summary of the LGM simulations with AWI-ESM. The paper is well written, the results are summarized clearly, and the underlying mechanisms are investigated thoroughly and convincingly. Below are a few minor comments:

**We really appreciate your positive comments and suggestions, which have significantly improved our manuscript. We have modified our paper accordingly. Details are as follows:**

L60, remove high-resolution (1.875 x 1.875 resolution is not high-resolution)

**Thanks for the comment, we now removed "high-resolution" from the text.**

L107, add "but not consistent with the estimate of -6.5 to -5.7 K cooling from Tierney et al., (2020).

**According to the reviewer's comment, we now completed the sentence, as following (please also refer to line 105-106 in the revised manuscript):**

*"Our full-forced LGM experiment simulates a mean global cooling by -4 K in comparison to PI, which meets the estimate based on various proxy reconstruction (Annan and Hargreaves, 2013; Annan et al., 2022), but not consistent with the estimate of -6.5 to -5.7 K cooling from Tierney et al. (2020). "*

L110, add values of hydrological sensitivity from Li et al., 2013 and Cao et al., 2019

**Yes, now in the revised version we changed the texts into (please also refer to line 108-111 in the revised manuscript):**
*"Thus the hydrological sensitivity of our model, defined as the slope between changes in global mean precipitation versus surface temperature, has the value of 1.625 %/K, which is relatively smaller as compared to other model studies. For example, Cao et al. (2019) yields a hydrological sensitivity of 2.2% K-1. According to Li et al. (2013), the values obtained from model ensemble mean are 2.64% /K and 2.28%/K for ocean and land, respectively."*

Fig.1, for each panel of Figure 1, it's helpful to add a corresponding scatter plot between the proxy temperature/precipitation and modeled temperature/precipitation at each location of the proxy site.

**Thanks for the comment, we now added three scatter plot of modeled versus reconstructed annual mean surface air temperature, precipitation, and sea surface temperature in updated Fig. 1 (at page 5).**

L156, please comment on the role of cold air from LGM ice sheet on the increase of AMOC

**Thanks for the comment, we now added in the updated manuscript (please also refer to line 160-162 in the revised manuscript):**
*"Moreover, the cold air transported from the glaciers is advected towards the subpolar region of the North Atlantic, resulting in an increase in seawater density in the upper layers. This, in turn, serves to further enhance the strength of the AMOC."*

L337, need to also mention that the orbital forcing is not expected to produce large JJA or DJF changes between LGM and PI since the precession parameters are similar.

**Thanks for the comment, we now added in the updated manuscript (please also refer to line 370-373 in the revised manuscript):**
*"Although the direct impact of orbital forcings on LGM temperature, precipitation, ENSO, and atmospheric circulation is considered negligible due to the small change in precession between the LGM and PI periods, it is important to acknowledge that Earth's orbital changes primarily influence the glacial climate indirectly through internal climatic feedback processes (e.g., via its long-term influences on the ice sheets and greenhouse gases)."*

**References**

Acosta Navarro, J. C., Ekman, A. M., Pausata, F. S., Lewinschal, A., Varma, V., Seland, Ø., Gauss, M., Iversen, T., Kirkevåg, A., Riipinen, I., et al.: Future response of temperature and precipitation to reduced aerosol emissions as compared with increased greenhouse gas concentrations, Journal of Climate, 30, 939–954, 2017.

Annan, J. and Hargreaves, J. C.: A new global reconstruction of temperature changes at the Last Glacial Maximum, Climate of the Past, 9, 367–376, 2013.

Annan, J., Hargreaves, J., and Mauritsen, T.: A new global climate reconstruction for the Last Glacial Maximum, Climate of the Past Discussions, pp. 1–20, 2022.

Boos, W. R. and Korty, R. L.: Regional energy budget control of the intertropical convergence zone and application to mid-Holocene rainfall, Nature Geoscience, 9, 892–897, 2016.

Cao, J., Wang, B., and Liu, J.: Attribution of the Last Glacial Maximum climate formation, Climate Dynamics, 53, 1661–1679, 2019.

D'Agostino, R., Bader, J., Bordoni, S., Ferreira, D., and Jungclaus, J.: Northern Hemisphere monsoon response to mid-Holocene orbital forcing and greenhouse gas-induced global warming, Geophysical Research Letters, 46, 1591–1601, 2019.

Donohoe, A., Marshall, J., Ferreira, D., and Mcgee, D.: The relationship between ITCZ location and cross-equatorial atmospheric heat transport: From the seasonal cycle to the Last Glacial Maximum, Journal of Climate, 26, 3597–3618, 2013.

D'Agostino, R. and Timmreck, C.: Sensitivity of regional monsoons to idealised equatorial volcanic eruption of different sulfur emission strengths, Environmental Research Letters, 17, 054 001, 2022.

Endo, H. and Kitoh, A.: Thermodynamic and dynamic effects on regional monsoon rainfall changes in a warmer climate, Geophysical Research Letters, 41, 1704–1711, 2014.

Frierson, D. M., Hwang, Y.-T., Fučkar, N. S., Seager, R., Kang, S. M., Donohoe, A., Maroon, E. A., Liu, X., and Battisti, D. S.: Contribution of ocean overturning circulation to tropical rainfall peak in the Northern Hemisphere, Nature Geoscience, 6, 940–944, 2013.

Han, Z., Zhang, Q., Li, Q., Feng, R., Haywood, A. M., Tindall, J. C., Hunter, S. J., Otto-Bliesner, B. L., Brady, E. C., Rosenbloom, N., et al.: Evaluating the large-scale hydrological cycle response within the Pliocene Model Intercomparison Project Phase 2 (PlioMIP2) ensemble, Climate of the Past, 17, 2537–2558, 2021.

Han, Z., Li, G., and Zhang, Q.: Changes in Sahel summer rainfall in a global warming climate: Contrasting the mid-Pliocene and future regional hydrological cycles, Climate Dynamics, 61, 1353–1370, 2023.

Held, I. M. and Soden, B. J.: Robust responses of the hydrological cycle to global warming, Journal of climate, 19, 5686–5699, 2006.

Hill, S. A., Bordoni, S., and Mitchell, J. L.: Axisymmetric constraints on cross-equatorial Hadley cell extent, Journal of the Atmospheric Sciences, 76, 1547–1564, 2019.

Hsu, P.-c., Li, T., Murakami, H., and Kitoh, A.: Future change of the global monsoon revealed from 19 CMIP5 models, Journal of Geophysical Research: Atmospheres, 118, 1247–1260, 2013.

Jacobson, T. W., Yang, W., Vecchi, G. A., and Horowitz, L. W.: Impact of volcanic aerosol hemispheric symmetry on Sahel rainfall, Climate Dynamics, 55, 1733–1758, 2020.

Kang, S. M., Held, I. M., Frierson, D. M., and Zhao, M.: The response of the ITCZ to extratropical thermal forcing: Idealized slab-ocean experiments with a GCM, Journal of Climate, 21, 3521–3532, 2008.

Kang, S. M., Frierson, D. M., and Held, I. M.: The tropical response to extratropical thermal forcing in an idealized GCM: The importance of radiative feedbacks and convective parameterization, Journal of the atmospheric sciences, 66, 2812–2827, 2009.

180     Lee, J.-Y. and Wang, B.: Future change of global monsoon in the CMIP5, Climate Dynamics, 42, 101–119, 2014.

Li, G., Harrison, S. P., Bartlein, P. J., Izumi, K., and Colin Prentice, I.: Precipitation scaling with temperature in warm and cold climates: an analysis of CMIP5 simulations, Geophysical Research Letters, 40, 4018–4024, 2013.

Li, T., Wang, Y., Wang, B., Ting, M., Ding, Y., Sun, Y., He, C., and Yang, G.: Distinctive South and East Asian monsoon circulation responses to global warming, Science Bulletin, 67, 762–770, 2022.

185     Lynch-Stieglitz, J.: The Atlantic meridional overturning circulation and abrupt climate change, Annual review of marine science, 9, 83–104, 2017.

Marshall, J., Donohoe, A., Ferreira, D., and McGee, D.: The ocean's role in setting the mean position of the Inter-Tropical Convergence Zone, Climate Dynamics, 42, 1967–1979, 2014.

McGee, D., Donohoe, A., Marshall, J., and Ferreira, D.: Changes in ITCZ location and cross-equatorial heat transport at the Last Glacial

190     Maximum, Heinrich Stadial 1, and the mid-Holocene, Earth and Planetary Science Letters, 390, 69–79, 2014.

Schneider, T., Bischoff, T., and Haug, G. H.: Migrations and dynamics of the intertropical convergence zone, Nature, 513, 45–53, 2014.

Tierney, J. E., Zhu, J., King, J., Malevich, S. B., Hakim, G. J., and Poulsen, C. J.: Glacial cooling and climate sensitivity revisited, Nature, 584, 569–573, 2020.